# The Feature Extraction through Wavelet Coefficients of Metal Friction Noise for Adhesive and Abrasive Wear Monitoring

**Yeonuk Seong, Donghyeon Lee, Jihye Yeom and Junhong Park \***

Department of Mechanical Engineering, Hanyang University, Wangsimni-ro 222, Seongdong-gu, Seoul 04763, Korea; tjddusdnr123@naver.com (Y.S.); alajju3@hanyang.ac.kr (D.L.); yom9417@hanyang.ac.kr (J.Y.)
\* Correspondence: parkj@hanyang.ac.kr; Tel.: +82-02-2220-0424

**Featured Application: Monitoring of Contact in Machining System for Possible Wears during Operation Movements.**

**Abstract:** Friction between metals is a physical phenomenon that occurs in manufacturing machine tools. This annoying noise implies unnecessary metal contact and deterioration of a mechanical system. In this study, for the monitoring of the friction between two metal surfaces, the acoustic signature was extracted by applying the wavelet transform method to the noise measured from the change in contact force for each state of adhesive and abrasive wear. Experiments were conducted with a constant relative speed between the contacting metal surfaces. For the adhesive wear, the peak signal-to-noise ratio (PSNR) calculated by the wavelet transformation increases with the increasing contact pressure. Opposite trends were observed for the abrasive wear. The proposed index formed a group within a specific range. This ratio exhibited a strong relationship with the wear characteristics and the surface condition. From the proposed index calculated by the wavelet coefficients, the continuous monitoring of the wear influence on the failure of the machine movement operations is achieved by the sound radiation from the contacting surfaces.

**Keywords:** friction; acoustic noise; adhesive wear; abrasive wear; wavelet coefficient



## 1. Introduction

For machine tools, moving components for material processing often come into contact with each other. Fatal and permanent wear occurs on the contacting surface by disintegration damage. We use lubricants, such as grease, to reduce wear by friction and provide the components with a smooth movement. Due to unexpected external influences, such as humidity and temperature variations, lubricants may not work properly, which causes severe frictional contact between metals. This frictional contact generates vibration and noise as well as heat, causing overload throughout the production process and eventually producing defective outputs or failures. The friction and wear of various materials on the profiles of surfaces have been conducted after considering the friction coefficient [1–4]. Ren et al. [4] studied friction behavior between the wafer and transfer film attached to each other. For a constant normal load, the friction coefficient gradually increased with the increasing sliding speed. For the ball-shaped chemical components, the friction coefficient was maintained in spite of high sliding speeds. Shih and Rigney [1] studied the variation of friction coefficients depending on the material coated on one or both sliding metal surfaces. The surface coated with magnetron tin film showed a smooth operation without much constraint. To investigate the friction of materials, the wear caused by friction always accompanies the vibrational excitation [5–7]. Chowdhury and Helali [5] studied the change of friction coefficients for various normal vibration amplitudes for materials of varying hardness. The friction coefficient decreased with the increasing vibration amplitude and converged to a specific value with the increasing rub cycle. To study the transient phenomenon, Stender et al. [8] studied the difference of some kinds of vibration by friction.

Conglin et al. [9] also investigated the frictional vibration and noise and influence of the transient friction characteristic. Many recent manufacturing machines require increased precision operations for applications to high-tech productions. In the manufacturing process, it is difficult to identify whether friction and wear occurred and whether maintenance operations are required.

Acoustical signature is a useful tool of application to continuously monitor production lines. The noise generation by friction and wear was a research subject for decades [10–12]. Abdelounis et al. [10] concentrated on the mechanism of sliding phenomenon on rough surfaces in terms of three steps (tribology, dynamics, and acoustics). The sound pressure level (SPL) was linear with surface roughness and sliding speed. Hase et al. [11] studied the characteristics of sounds from abrasive and adhesive wears in frequency domain. In terms of amplitude and frequency, the adhesive wear generated sounds in various frequency bands. For the abrasive wear, it was single but had a wide frequency range. Regardless of the type of materials used in the experiments, it maintained consistency in its generation frequency spectrum range. There was a transition period between adhesion and abrasive wears. The frequency bands of sound radiation from contacting surfaces are different. Therefore, through a series of signal processing, it was necessary to classify the sliding mechanism and derive the factors for the occurrence of wear. It was possible to identify whether the current state is influenced by friction or not. Friction noise based on frequency domain was analyzed using fast Fourier transform (FFT) with statistical calculations [12–16]. Le Bot [13] studied the power spectral density (PSD) of sounds for various roughness of surfaces and relative motion speed of materials. Aguilar et al. [14] evaluated surface roughness from friction noise, and a statistical index was used for artificial neural network (ANN). From the index feature, different roughnesses of surfaces were well discriminated with the ANN algorithm. Akay [17] reviewed the characteristics of friction phenomenon for various mechanical systems as a standard to investigate the motion of sliding surfaces. Recently, there were some studies related to extracting features to diagnose or detect components of fault in mechanical systems.

As the number of mechanical parts is increasing in production, it is becoming more difficult for operators to make judgements regarding maintenance. A self-diagnosis system that relies on sensor measurements is essential. For production with minimal output of defective parts, interference of tools or parts should be monitored in real-time. For this reason, research on signal processing methods for the self-diagnosis of machines and the use of artificial intelligence (AI) has been expanding recently. Simon and Deivanathan [18] used the vibration response and extracted features to detect the faults. Ahmed et al. [19] showed the proper application of the monitoring system in machining centers by using acoustic emission responses. Li et al. [20] used empirical wavelet transform to conform denoising of vibration responses. They focused on machining components or tools and treated the response or FFT data with statistical analysis. There is a limitation on analyzing both in time and frequency domains. It is required to analyze transient variations of the spectral characteristics to analyze frictional sounds. In this study, the feature extraction method from the acoustic response is used to investigate the adhesive and abrasive wears between two metal materials. The extraction of features of friction sounds with variation of the contact pressure was reflected by the proposition of a system capable of self-diagnosis.

## 2. Wavelet Transform for Sounds from Frictional Contacts

### 2.1. Adhesive and Abrasive Wears

There are four wear modes caused by friction between two metal components (adhesive, abrasive, fatigue, and corrosive) [21–23]. Adhesive, abrasive, and fatigue wears occur by mechanical contacts. The adhesive and abrasive friction wears generate in a relatively short time. The correlation between the amount of loss due to wear and the surface conditions were studied in previous studies [24,25]. In this study, adhesive and abrasive wears were monitored. When two metal components are in strong contact with each other, the unexpected large contacting force can cause permanent deformation to each other.

The mechanism is distinguished by the cause of its occurrence. Figure 1a,b show the process of adhesive and abrasive wears, respectively. Under relative motion, one component scratches with high pressure by generating heat caused by dislocations [21,23]. In order to simulate the metal contact, a system that imparts frictional contact between two metal components through a reciprocating motion was constructed, as shown in Figure 2. With the compressed force applied between the two parts, damage occurs along the slip plane due to tangential shear. This mode corresponds to adhesive wear. The abrasive wear occurs when relative motion occurs due to different material hardness. For a metal of large hardness, it is generally more susceptible to abrasive wear. When small particles repeatedly scratch the surface by relative motion between two flat components, they form a groove on the surface as the abrasive wear develops.

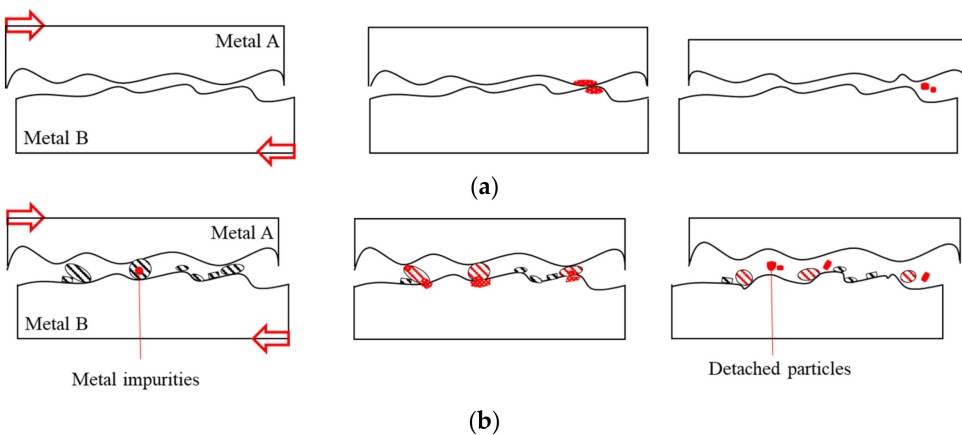

**Figure 1.** Mechanisms of two different wears: (**a**) adhesive wear and (**b**) abrasive wear.

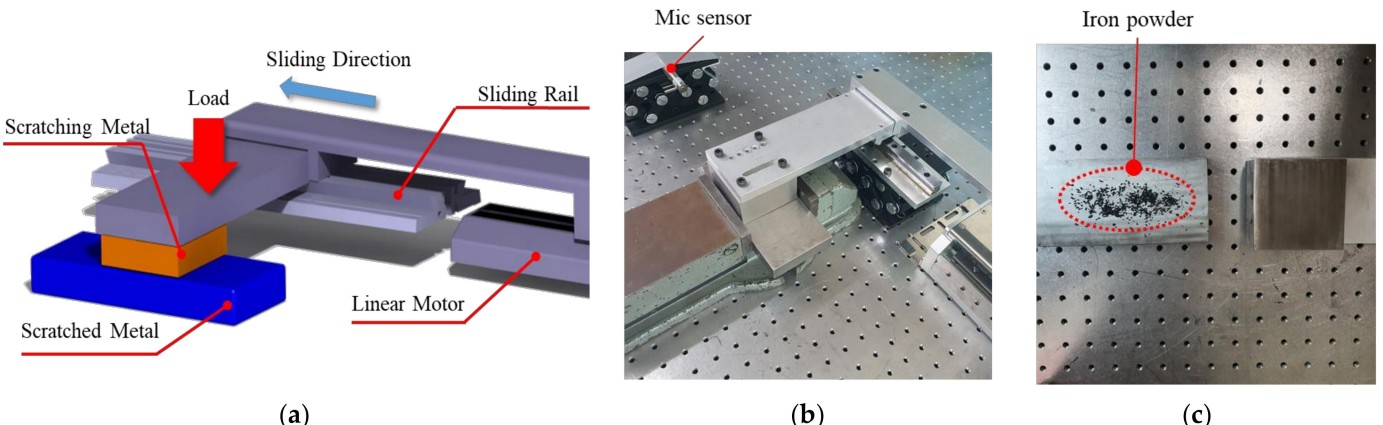

**Figure 2.** Experimental setting for scratching on the metal for adhesive and abrasive friction: (**a**) schematic section view of scratching experiments using linear motor; (**b**) metal friction using linear motor; (**c**) iron powder on metal to realize abrasive wear.

### 2.2. Haar Wavelet Coefficient and PSNR

The instantaneous motion caused by friction between two components induces plastic deformation of the surface and generates acoustic energy and heat. Noise from the frictional wear generates as the local response rather than the modal vibration responses. The contact pressure and sliding velocity have significant influence on the response. The frequency spectrum variation at regular intervals requires monitoring for the inspection of the rapid time transition section of the noise caused by friction. Because of the trade-off relationship between time and frequency intervals in short-time Fourier transform (STFT), it is not straight forward to extract components from the response due to wear in a short time.

The experiments were performed at the same relative speed of 50 mm/s. During the experiments, care was given to replace the mating surfaces to minimize the influence of wear by the previous measurements. By adjusting the height of the jig, the contact pressure between the two metals was varied. Measurements were conducted for three different contact pressures of 5, 11, and 23 kPa. This contact pressure was measured using the load cell (Model KMR-20 kN, HBM Inc., Darmstadt, Germany), which is set on the scratching metal block. All cases were performed under dry friction. In the experiment, the sound radiation was acquired with a sampling rate of 131,072 Hz by using a data acquisition device (Model 3560-B-030, Brüel & Kjær Inc., Nærum, Denmark). System response was acquired by using a microphone sensor (Model 378B02, PCB Piezotronics Inc., New York, NY, United States), and it was installed and measured at the same height 10 cm away from the scratch. To confirm the reproducibility of the experiment, the same signal processing procedure described above was applied to a friction signal of 1 s at different data points. Wavelet transform (WT) and PSNR calculations were performed on the acquired time data using MATLAB.

Figure 3a shows the sound response measured by the microphone for the two metals, as represented in Figure 2b. Since the friction randomly varied in magnitude, it was difficult to capture consistent statistical characteristics. Figure 3b shows the STFT results. The frequency distribution of the acoustic energy changed continuously and irregularly with time. Despite this, there is a section that showed a very high response of up to 2500 Hz; this frequency response does not contain the contribution from friction components. It was generated from the vibrations of two metals where scratches occurred. This makes it difficult to derive the friction components. This variability is a major obstacle in extracting the inherent characteristics of friction when monitoring a metal system. The process of decomposing the signal for each band was efficient.

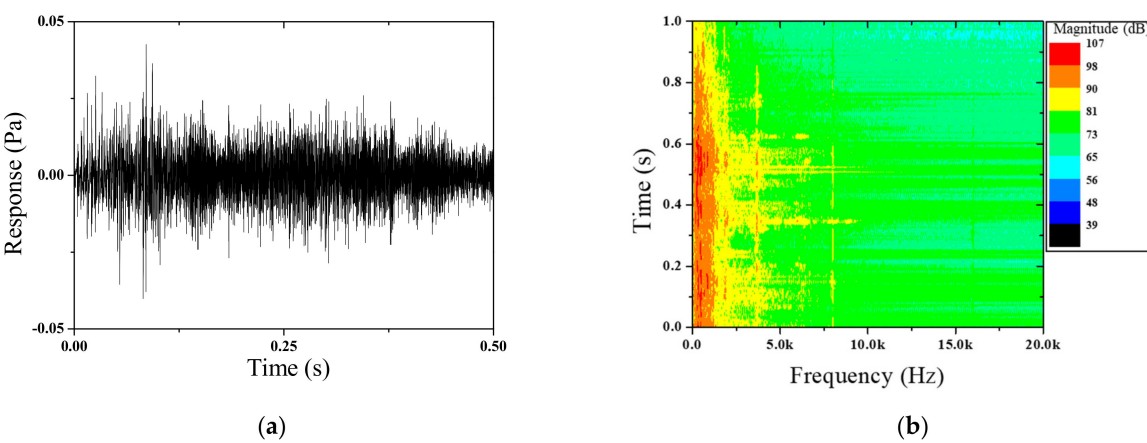

(**a**)　　　　　　　　　　　　　　　　　　　　　　　(**b**)

**Figure 3.** Sound responses from the metal contact surface: (**a**) transient response and (**b**) STFT result.

WT complements the shortcomings of this STFT. In particular, Haar-WT is a convenient method for analyzing acoustic signals expressed in time. In addition, it shows good performance for the decomposition of the signal with the average change of magnitudes [26–28]. Haar-WT distinguishes detail and coarse coefficients, which is calculated from the original response, *y*, as represented by [29]:

$$d_k = \alpha(y_{2k} - y_{2k-1}), \; c_k = \alpha(y_{2k} + y_{2k-1}),\tag{1}$$

where subscription $k$ represents the order of acoustic response samples. The $c$ and $d$ denote the coarse and detail components, respectively. Wavelet constant $\alpha$ is always $1/\sqrt{2}$ because of energy conservation for normalizing:

$$d_k^2 + c_k^2 = 2\alpha^2 \left( y_{2k}^2 + y_{2k-1}^2 \right).\tag{2}$$

For the Haar-WT in fine-scale detail and approximation, the consecutive process of the calculation of Equation (1) is performed. In this way, the approximated response in level $j$ is extracted by accompanying the scaling function $\phi$ and wavelet function $\psi$ with detail and coarse coefficients [30]:

$$y_{j+1}(t) = \sum_{k=0}^{2^j-1} \left\{ c_{j,k}\phi_{j,k}(t) + d_{j,k}\psi_{j,k}(t) \right\}, \tag{3}$$

where the $\phi$ and $\psi$ also represent low-pass filter and high-pass filter components, respectively, which is calculated as

$$\phi_{j,k}(t) = 2^{j/2}\phi\left(2^jt - k\right), \phi(t) = \begin{cases} 1 & \text{if } t \in [0,1] \\ 0 & \text{otherwise} \end{cases}, \tag{4}$$

$$\psi_{j,k}(t) = 2^{j/2}\psi\left(2^jt - k\right), \psi(t) = \begin{cases} 1 & \text{if } t \in [0,1/2) \\ -1 & \text{if } t \in [1/2,1) \\ 0 & \text{otherwise} \end{cases}. \tag{5}$$

From Equation (3), detail and coarse coefficients in the level $j - 1$ can be extracted by Euler formulas as

$$c_{j-1,k} = \int_{2^{-j-1}(k)}^{2^{-j-1}(k+1)} y(t)\phi_{j-1,k}(t)dt, \tag{6}$$

$$d_{j-1,k} = \int_{2^{-j-1}(k)}^{2^{-j-1}(k+1)} y(t)\psi_{j-1,k}(t)dt. \tag{7}$$

Here, the wavelet function is denoted as $\psi(t) = \phi(2t) - \phi(2t - 1)$. The wavelet coefficient for coarse and detail components can be briefly extracted by rearranging Equations (6) and (7) with the following relation:

$$c_{j+1,k} = \alpha\left(c_{j,2k} + c_{j,2k+1}\right), \; d_{j+1,k} = \alpha\left(c_{j,2k} - c_{j,2k+1}\right). \tag{8}$$

We can evaluate the detail and coarse coefficients in level $j + 1$ from the coefficients in level $j$. By repeating the procedures, we can continuously decompose the coarse coefficient as the level processes. In the last level, the arrangement of the coefficient consists mostly of detail coefficients. Figure 4 shows the progress of the discrete wavelet transform. Each level reflects the frequency band. For $n$ sampling frequency in level $j$, coarse and detail coefficients reflect the frequency band from 0 Hz to $n/(j + 1)$ Hz and from $n/(j + 1)$ Hz to $n$ Hz, respectively.

The change in the period of the detail coefficients reflects the variation in the high-frequency components that is involuntarily induced by the frictional wear. Regarding the wavelet coefficient in the high-frequency range, it is advantageous for it to be expressed as the single value of change in response to the contact phenomenon between the components. This PSNR is expressed as below [31,32]

$$P = 10\log_{10}^{(d_{peak}^2/E)}, \tag{9}$$

where $d_{\text{peak}}$ denotes the detail coefficient peak value, and $E$ is the mean square error of the coefficient array. The $E$ is computed as

$$E = \frac{1}{n}\sum_{i=1}^{n}\left(d_j - \hat{d}_j\right)^2. \tag{10}$$

Here, $\hat{d}_j$ denotes the predicted value calculated by a regression model based on given wavelet coefficients. The $E$ represents how far the data are located from the average values. The PSNR statistically represents the fine change in the occurrence of the frictional wear.

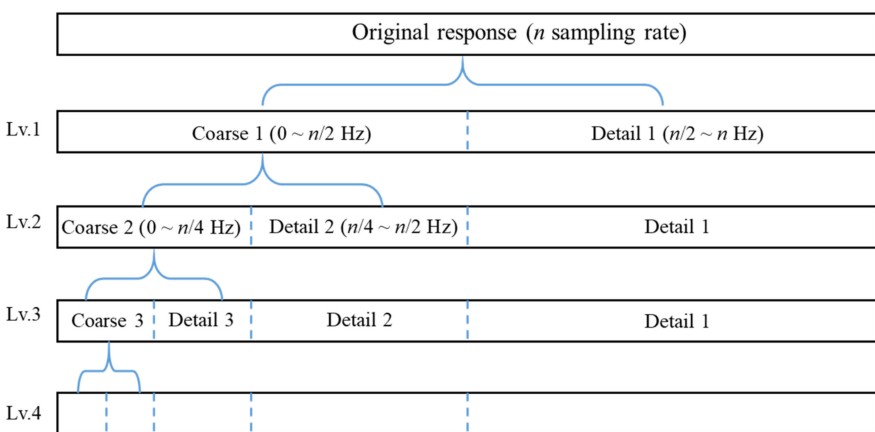

**Figure 4.** Process of wavelet decomposition in the primary stage for original acoustic response.

### 2.3. Specimen and Experimental Setting

In this study, the constituting components where friction occurs were made of steel to simulate the frictional wear in machine parts. The parts, such as the linear motion guide or main shaft, consisted of steels. Relative velocity of the guide part was 30 to 100 mm/s. To consider the sliding situation, two flat metal slides were facing each other, as configured in Figure 2. The contact pressure was assumed to act without lubricant. With the action of the contact pressure, the noise was generated from the relative motions. The metal jig was fixed to the linear motor (PARKER Inc., Mayfield Heights, OH, USA) to perform stroke motions for implementation of a constant relative velocity between the contacting surfaces. Figure 2a shows an overview of the experimental setup for the rubbing of the two metals. The change in contact pressure between two metal components was adjusted under various heights. To realize the adhesive wear, metal moves along the other surface. This movement generates the heat and noise from friction. For metal processing, large metal wastes are removed without difficulty. There is a possibility of small pieces attaching to the surface of the motion guides. These attachments induce the abrasive wear. In order to realize the abrasive wear, a little iron powder oxidized by heat with an average diameter of 1.5 mm was sprinkled between the two metals and distributed evenly to induce a steady motion. All friction-component surfaces have a roughness ($R_a$) of 1.2 μm. Figure 2c shows the iron powder, which represents the impurity components in temporal moving systems.

## 3. Wavelet Coefficient and Feature Extraction

### 3.1. Effect of Wears on Wavelet Coeffcients of Radiate Sounds

Discrete wavelet coefficient was calculated for specimens influenced by the adhesive wear, as shown in Figure 5a. As the wavelet number increased, the obtained coefficient increased gradually. A small coefficient value indicates a high frequency. The periodicity of the coefficient value changed in the second level, as shown in Figure 5b. Particularly, the frequency band corresponding to level 2 includes both the adhesive and abrasive friction noise ranges [11]. For minimal pressure applications, there was no significant change in periodicity in the second level. As the contact pressure increased, the sound level increased. The high-frequency components for scratching caused by contact pressure appeared in the detail coefficients, especially in level 2. Figure 6a,b show the wavelet coefficients for the abrasive wear. The effects of the wear appeared at the second level of the detail coefficients. The abrasive wear did not have a significant difference in the detail coefficient values compared to those of the adhesive wear. The cycle change of coefficient values was lower. The noise tended to a specific signal and can be compared using PSNR.

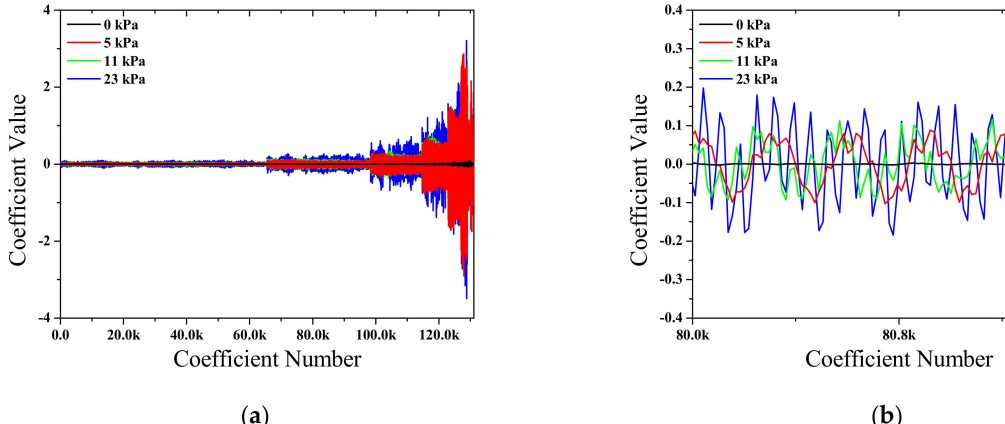

**Figure 5.** Wavelet coefficient for the detail responses and influence of the contact pressure for adhesive wear: (**a**) detail wavelet coefficients in overall range of 1 s and (**b**) detail wavelet coefficients in level 2.

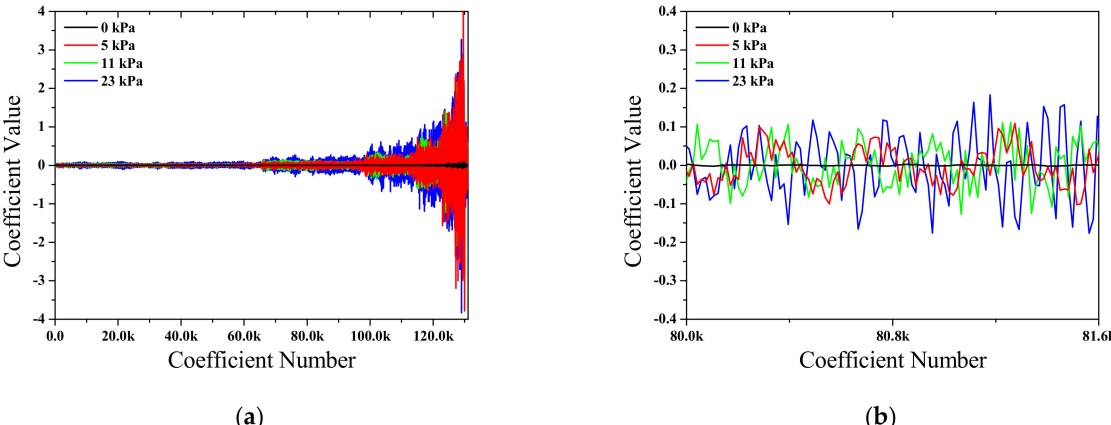

**Figure 6.** Wavelet coefficient for the detail responses and influence of the contact pressure for abrasive wear: (**a**) detail wavelet coefficients in overall range of 1 s and (**b**) detail wavelet coefficients in level 2.

### 3.2. Feature Extraction from PSNR

PSNR values were evaluated through Equation (9) for the measured responses. For the adhesive wear, PSNR increased as the contact pressure increased, as was observed in the measured sound pressure in Figure 7a, but there was no tendency for level 1 as expected. The surface irregularity was maintained due to adhesion phenomenon, even if the relative motion occurred continuously. In addition, as the contact pressure increased, the adhesion surface area increased, which caused more acoustic noise. On the other hand, in the case of abrasive wear, the PSNR tended to decrease as the contact pressure increased. The PSNR evaluated from the detail coefficient in abrasive wear is shown in Figure 7b. The calculated value from level 1 did not show any consistent variation. For the abrasive wear, the surface was evenly distributed due to friction by the impurity particles. The contact pressure increased the distribution. Under the adhesive wear, the fracture and crack on the surface of the metals influenced the dynamic responses. With the increasing pressure, the crack and fracture progress increased. This induced a larger noise ratio in the acoustic responses. On the other hand, there should be much more impurities on the surface under abrasive wear. The flat surface did not radiate a larger noise. With the increasing pressure, the degree of the abrasive wear progressed faster and caused smaller cracking of the surface. It was confirmed that the frequency component variation from scratching affected the period of the wavelet coefficients. It can be seen that the emission period of the acoustic energy varied depending on the contact pressure and was successfully classified as the PSNR. Figure 8 shows the results of the tendency and reproducibility of the PSNR values according to the

contact pressure at level 2 for adhesive and abrasive wear. In the case of both adhesive and abrasive wear, it was confirmed that the PSNR values formed a group, and the tendency was reversed. This group formation can be used as a monitoring factor in the system self-diagnosis using acoustic sensors.

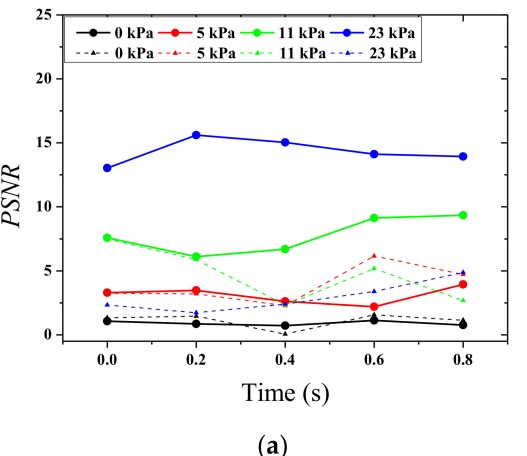 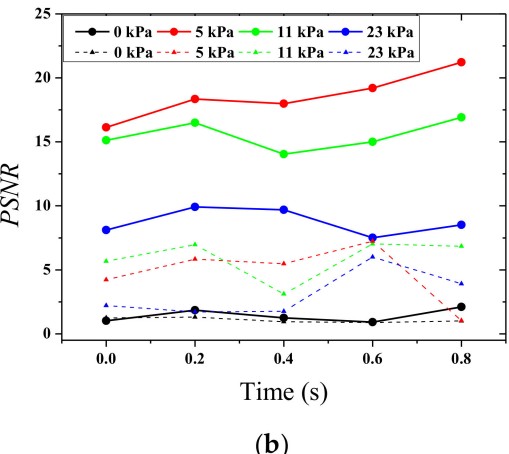

(**a**) (**b**)

**Figure 7.** PSNR for various time intervals calculated from the detail wavelet coefficients. The solid line (-) denotes the level 2 data and the dash line (—) denotes the level 1 data: (**a**) the adhesive wear and (**b**) the abrasive wear.

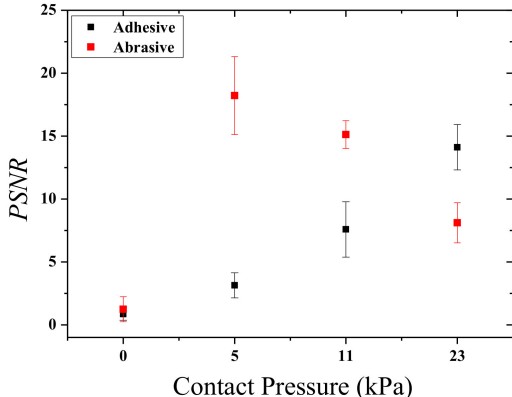

**Figure 8.** PSNR tendency according to contact pressure for two friction mechanisms.

## 4. Discussion

In this study, friction noise from the relative motion of metals was analyzed in order to detect unnecessary contacts and wears that may occur in a mechanical sliding system. The proposed methodology enhances the monitoring of the contact status. It was efficient in deriving periodic changes in the wavelet coefficient by using the WT method. We considered contact pressure to index the wear feature. The magnitude of the contact pressure was relatively well identified. A distinct difference was observed in the periodicity in a specific wavelet number section. It expressed the wear magnitude by the index using statistical calculations. The PSNR increased with the increasing pressure for the adhesive wear. The opposite variation was observed for the abrasive wear. As the contact pressure increased, two different wear mechanisms occurring on the surface were well shown by the statistical analysis of the noise. The reliability of the wavelet coefficient was validated by the experimentations. Further signal processing methodologies are required for effective separation of where the adhesive and abrasive wears overlap.

## 5. Conclusions

This study presents the detection of metallic contact by using the acoustic signal. From the Haar wavelet coefficients derived by the discrete wavelet transform, it was possible to determine whether the friction was caused by the adhesive and abrasive wears. Through the PSNR, the influence of the contact pressure was identified. This grouping factor value serves as an indicator for the real-time monitoring of industrial machinery in the manufacturing process. Particularly, the development of the abrasive or adhesive wears evaluated via the proposed process. This study is expected to be sufficient as an academic basis for diagnosing the state of mechanical systems when used for relatively simple processes and equipment.

**Author Contributions:** This paper is a scientific manuscript incorporating contributions from all authors. Conceptualization, methodology, validation, and software Y.S.; investigation D.L. and J.Y.; writing (draft, review, and editing) Y.S.; supervision and administration, J.P.; funding acquisition, J.P. All authors have read and agreed to the published version of the manuscript.

**Funding:** This research was supported by the Korea Institute of Energy Technology Evaluation and Planning (KETEP) grant funded by the Korea government (MOTIE) (20206300000070, Development of Intelligent Monitoring/Diagnosis Technology for IoT-based Re-Manufacturing Industry Machines).

**Institutional Review Board Statement:** Not applicable.

**Informed Consent Statement:** Not applicable.

**Conflicts of Interest:** The authors declare no conflict of interest.

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
