# Peer review of "The Feature Extraction through Wavelet Coefficients of Metal Friction Noise for Adhesive and Abrasive Wear Monitoring"

_applsci, doi:10.3390/app11093755_

Round 1

Reviewer 1 Report

The authors present an interesting methodology for adhesive and abrasive wear monitoring in sliding friction through feature extraction. An overall interesting and comprehensible paper.

However, a few questions remain open and improvements are suggested.

Line 63: dot missing after et al.
Line 113: Abbreviation STFT was not introduced and explained
Line 122: unit missing after the number (1/s)
Line 191: Effect of wears cannot be plural
Line 195: the sentence is not comprehensible (The change in the frequency spectrum with to the contact pressure...
Line 197: adhesive as an adjective
Figure 6 caption should be completed

The authors suggest the methodology for the monitoring of sliding parts in machine tools. However, the analogy case was not explained. Which moving parts of machine tools are represented with the test setup? How are the parameters like sliding speed, contact normal stress motivated? How can the results be used for the monitoring of machine tools in practice?

Adhesive wear strongly depends on the material specifications. Which metals have been used as sliding partners? Was any wear observed in the experiment?

Abrasive wear was modeled as three-body abrasion using iron powder. Please give more specifications on the used powder (iron oxide? diameter) and the quantification of wear.

Reviewer 2 Report

In manuscript, the authors studied the acoustic signature created by friction between two metal surfaces. This acoustic signature was extracted by applying the wavelet transform method to the noise measured with the change in contact force for each state of adhesive and abrasive wears.

The paper is interesting, but needs some corrections. The following comments should be addressed before publishing in Applied Sciences.

  1. The literature review should be modified to include the latest literature. In paper, the bibliography is from several or a dozen years ago. Recent literature is missing. Only two from 2019 and one from 2017 were cited in the paper.
  2. Please divide the chapter Discussion and Conclusion into two parts. Both the discussion of the results and the conclusions should be presented in a separate section.

Reviewer 3 Report

The research is usually clearly formulated. Some ambiguities in the claims are included in the comments. Authors should highlight the profound benefits of research compared to similar research by other authors so that paper could be accepted.

Remark 1:

Line 11 -,,This annoying high-frequency noise implies imperfect metal contacts in the testing system.“

Can this statement be justified? In the case of perfect contact, cannot this phenomenon occur? This strong statement should be explained better. What does it mean imperfect contact in this problem?

Remark 2:

Line 15 – „Peak Signal-to-noise“ change to peak signal-to-noise

Remark 3:

Line 72 – “Fourier transforms form alone have a limitation in its effectiveness due to the trade-off relationship between frequency and time resolutions.” – Since authors are using other signal processing methods (Wavelet transform), this statement should be justified in more detail. What are the limitations of FFT in terms of wear detection from the acoustic signal?

Remark 4:

Line 113 abbreviation STFT wasn’t previously defined. Short-time Fourier transform?

Remark 5:

Line 127 – “WT” abbreviation wasn’t defined yet. It is defined only on line 143 for the first time.

Remark 6:

Line 172 – Formula for error “E” in equation (9) should be introduced or at least a closer description of error should be presented in the text.

Remark 7:

Line 234 – “b) adhesive” perhaps it should be “abrasive”

Remark 8:

Chapter 2.2 - Why Haar wavelet form was used? Is it the most effective wavelet form for this analysis? If so, it could be mentioned in the text.

Remark 9:

Line 193 „The small values from the left show the frequency band with the highest resolution. As the number of coefficients increased, the wavelet coefficient of the lower frequency band was obtained.” This statement is unclear for the reader who is not working daily with Wavelet Transformations. Perhaps the increase of the coefficient values over time could be explained differently. 

Remark 10:

In the discussion and conclusion, some comparisons with similar research and added value should be emphasized.

Round 2

Reviewer 3 Report

All comments were incorporated and explained by the authors. I therefore recommend the article for publication.